# Effect of BIO-PLY^TM^, a Platelet-Rich Plasma Derived Biologic on PRRSV-2-Infected Macrophages

**DOI:** 10.3390/v14122666

**Published:** 2022-11-28

**Authors:** Alba Frias-De-Diego, Jessica M. Gilbertie, Frank Scholle, Sarah Dejarnette, Elisa Crisci

**Affiliations:** 1Department of Population Health and Pathobiology, College of Veterinary Medicine, North Carolina State University, Raleigh, NC 27607, USA; 2Department of Biomedical Affairs and Research, Edward Via College of Osteopathic Medicine, Blacksburg, VA 24060, USA; 3Department of Biological Sciences, North Carolina State University, Raleigh, NC 27695, USA

**Keywords:** antivirals, biologic, porcine macrophages, PRRSV

## Abstract

Porcine Reproductive and Respiratory Syndrome (PRRS) is the one of the most devastating diseases impacting the swine industry worldwide. Control and prevention methods rely on biosafety measures and vaccination. As an RNA virus with a high rate of mutation, vaccines are only partially effective against circulating and newly emerging strains. To reduce the burden of this disease, research on alternative control methods is needed. Here, we assess the in vitro antiviral effect of a novel platelet-rich plasma-derived biologic termed BIO-PLY^TM^ (for the BIOactive fraction of Platelet-rich plasma LYsate) from both swine and equine origin. Our results show that BIO-PLY^TM^ significantly reduces the amount of PRRSV viral load determined by RT-qPCR and the number of infectious viral particles measured by TCID50 in infected porcine alveolar and parenchymal macrophages. This study also showed limited toxicity of BIO-PLY^TM^ in vitro and aspects of its immunomodulatory capacity evaluating the regulation of reactive oxygen species and cytokines production in infected cells. Finally, this study presents promising data on the effect of BIO-PLY^TM^ on other RNA viruses such as human A influenza viruses and coronavirus.

## 1. Introduction

In 2019, international commerce of livestock and swine products surpassed USD 20 billion worldwide, and over USD 7 billion in the U.S. alone as reported by the United States Department of Agriculture (USDA) [1]. Porcine Reproductive and Respiratory Syndrome virus (PRRSV) is one of the most prominent diseases worldwide in the swine industry, causing over USD 600 million in losses every year in the U.S. [2,3]. PRRSV is an enveloped positive single stranded RNA virus with tropism for the cells from the monocytic lineage, particularly macrophages that are known to be the virus target cell [4]. PRRSV is divided into two species: PRRSV-1, mainly present in Asia and Europe, and PRRSV-2, with higher prevalence in the U.S. [4]. As many other RNA viruses, PRRSV displays a high mutation rate and ability to adapt and spread rapidly within and between herds [5,6]. The current control and prevention measures for PRRSV infection are based on biosafety protocols and immunization [7]. However, given the nature of the virus and its high mutation rate [5], approximately 20–30% of US breeding herds still become infected with PRRSV every year, since vaccines cannot yet achieve complete cross protection to circulating and newly emerging strains [8]. For that reason, research on alternative treatments that could lead to a reduction of the burden of this disease are of interest. Some approaches that have proven their effectiveness against PRRSV in vitro are plant extracts like *Caesalpinia sappan* [9], tea polyphenols [10], nanoparticles [11], carbon-based materials [12] or mycotoxins [13] among others. However, a number of these options, such as nanoparticles [11] or carbon-based materials, [12] are either difficult or costly to obtain or can lead to secondary effects when used long-term [13]. Therefore, finding an alternative that can be used in swine industry that surpasses those obstacles while maintaining low production costs is the key for a successful and applicable product.

Platelets play an important role in innate immunity since they degranulate and release antimicrobial peptides (AMPs) when activated [14,15]. Platelet rich plasma (PRP) is an autologous compound obtained from an individual’s blood with numerous applications within the biomedical field [16,17,18,19,20,21,22,23,24]. Originally, PRP was found to be an effective treatment for musculoskeletal conditions such as osteoarthritis [25,26,27] or bone infections [28], among others [16,17,18,19,20,21,22,23,24]. As one of the main complications of traumatic lesions is potential subsequent bacterial infections [29,30,31], the discovery of the antimicrobial capacities of PRP brought great advantages to its utilization, since the application of a single compound could aid with multiple complications at a time such as pain relief [18], immune regulation [24] and bacterial control [27,32,33,34,35]. Recently, the antiviral properties of PRP have been described for SARS-CoV-2 [36,37], leading the way for PRP and its derivatives to be applied as safe and effective antivirals.

Along with the current interest in PRP utilization, the antimicrobial properties of the acellular component of PRP led to the use of PRP lysate (PRP-L), that presented a higher concentration of platelet bioactive factors than the original PRP [38,39]. This change opened the possibility for the allogeneic use of PRP-L [38], along with the ability to pool PRP-L from multiple individuals and thus decrease variability in the final product [40,41]. Our group has recently described a novel processing method that isolates the AMPs within PRP and has provided extensive in vitro [32] and in vivo [27] data supporting its antibacterial activity. The resulting product from this processing method is the small molecular weight, cationic peptide fraction of pooled PRP-L and is termed BIO-PLY^TM^ for the BIOactive fraction of Platelet-rich plasma Lysate, (U.S. Patent Application #20210128626) [33].

The objective of this study was to test the antiviral properties of porcine- and equine-derived BIO-PLY^TM^ against three strains of PRRSV-2: a high pathogenic (NC-174), a low pathogenic (NC-134) and a reference strain (VR-2332) during infection in porcine alveolar macrophages (PAM) and lung pulmonary intravascular macrophages, designated as PIM. Particularly, this study focuses on the ability of swine and equine BIO-PLY^TM^ to reduce viral load and to modulate cytokine and reactive oxygen species (ROS) production in infected macrophages in vitro. Finally, this study also shows promising preliminary data on the use of inter-specific BIO-PLY^TM^ and its effect on other RNA viruses such as human influenza and coronavirus.

## 2. Materials and Methods

### 2.1. Cells

Monkey African Green kidney (MA-104) cell line (ATCC, CRL-2378.1T), Madin-Darby Canine Kidney cells (MDCK, BEI NR-2628) and Newborn Pig Trachea cells (NTPr) were cultured in Dulbecco’s Modified Eagle Medium (DMEM) supplemented with 10% inactivated fetal bovine serum (FBS), 100 IU/mL penicillin, 100 mg/mL streptomycin, 2 mM L-glutamine at 37 °C in a 5% CO_2_ atmosphere.

Porcine alveolar macrophages (PAM) and parenchymal macrophages, including pulmonary intravascular macrophages (PIM), were isolated from lungs of *Mycoplasma*, influenza A virus and PRRSV-negative pigs as previously described [42,43]. Briefly, PAM were obtained through broncho alveolar lavage using PBS supplemented with penicillin-streptomycin (GenClone, San Diego, CA, USA). Then, cells were centrifuged, resuspended in RPMI media, and counted. To obtain PIM, the parenchymal region of the lung was collected and digested using a combination of DNAse (0.1 mg/mL), collagenase (2 mg/mL) and dispase (1 mg/mL). Lung tissue fragments were crushed and filtered using 100 μm cell strainers prior to the treatment with Red Blood Lysis Buffer following manufacturer’s instructions (ChemCruz, Santa Cruz, CA, USA). Finally, cells were stored as parenchymal cells (PAR) at −80 °C. Prior to each experiment, lung mononuclear phagocytes (MNP) that include macrophages and different subsets of dendritic cells (DC) were isolated from PAR by a 1.065 density iodixanol gradient using OptiPrep (Stem Cell, Vancouver, BC, Canada) as previously described [42].

### 2.2. Macrophage Isolation

To further separate macrophages from the rest of MNP pool, two different methods were compared: (1) through magnetic bead separation using EasySep™ PE Positive Selection Kit II (Stemcell, Vancouver, BC, Canada); (2) using the capacity of macrophages to attach to a plastic culture plate to separate them from the non-adherent MNP that remain in the supernatant [44,45].

For the magnetic bead separation, enriched MNP were incubated with the provided selection mix containing anti-mouse IgG1 and mouse IgG1 anti-pig CD163 monoclonal antibody (Bio-Rad, MCA2311GA, IgG1 clone BL1H7) at 1 mg/mL for 15 min at room temperature (RT) following manufacturer’s indications. After separation, macrophages were resuspended in RPMI medium supplemented with 10% inactivated fetal bovine serum (FBS), 100 IU/mL penicillin, 100 mg/mL streptomycin, 2 mM L-glutamine, referred to as complete RPMI (RPMIc).

For the isolation via attachment, enriched MNP were seeded in a culture plate and incubated for 2 h at 37 °C in a 5% CO_2_ in RPMIc. After incubation, the supernatant containing the non-adhered MNP was removed [45].

### 2.3. Viruses

For this study, two PRRSV-2 field isolates were used, the high pathogenic NC-174 (NC-174G, GenBank accession ID ON844089) and the low pathogenic NC-134 (GenBank accession ID ON844087). Additionally, the reference strain VR-2332 (ATCC, vr-2332) (GenBank accession ID AY150564.1) was used as a standard for comparison. Virus stocks were expanded using cultures of PAM and MA-104 cell lines as previously described [46] and stored in aliquots at −80 °C. For additional analyses, we used human influenza A from swine origin (Influenza A Virus, A/California/04/2009 (H1N1) pdm09, BEI NR-13659) and Influenza A virus A/Wisconsin/67/2005 (H3N2), BEI NR-9694) and human coronavirus (HuCoV229E).

### 2.4. BIO-PLY^TM^ Production

BIO-PLY^TM^ was produced as previously described [27,33]. Briefly, whole blood from either healthy horses or *Mycoplasma*, influenza virus and PRRSV-free pigs from NCSU herds was collected in vacutainer tubes containing the anticoagulant acid citrate dextrose (ACD). The Institutional Animal Care and Use Committee of North Carolina State University approved this protocol. Erythrocytes were allowed to settle at RT for 30 min using the addition of 3% dextran to whole blood to aid in sedimentation, and the layer above the erythrocytes containing leukocytes, platelets and plasma known as leukocyte-rich platelet-rich plasma or L-PRP was collected and centrifuged at 300× *g* for 15 min to remove leukocytes. The pellet was then discarded and the supernatant containing the platelets and plasma was centrifuged at 1500 g for 15 min. The supernatant containing the platelet-poor plasma (PPP) was saved, and the pelleted platelets were resuspended in PPP to obtain a platelet-rich plasma (PRP) of an approximate concentration of 50× that of whole blood. The 50× PRP, defined as containing greater than 10,000,000 platelets/µL, <100 WBC/µL and >10 RBC/µL. PRP lysate, (PRP-L), was generated by bead homogenization [27,32,33,47].

Cell debris from lysis was removed through centrifugation at 20,000× *g* for 20 min before a final filtration. To remove anionic components, PRP-L was incubated with a washed, loose anion exchange resin (UNOsphere Q resin, Bio-Rad Laboratories, Hercules, CA, USA), followed by a separation of unbound components (cationic and neutral) using a bottle-top filter (0.22 µm PES, Nalgene, ThermoFisher, New York, NY, USA). Fractionation by molecular weight was performed with a 10 kDa molecular weight cutoff filter (Amicon^®^ Ultra 15 mL Centrifugal Filters, 10 kDa, Millipore Sigma, Burlington, MA, USA). The final swine or equine filtrate (BIO-PLY^TM^) containing proteins and peptides <10 kDa in size was collected, aliquoted into 1 mL aliquots, and stored at −80 °C until use in this study.

### 2.5. Experimental Layout

Infection timepoints, multiplicity of infection (MOI), BIO-PLY^TM^ concentration and treatment settings were first standardized using the MA-104 cell line prior to primary cell infection (Figure 1). Tested timepoints were 12 h and 24 h and evaluated MOIs were 1, 0.5 and 0.05. BIO-PLY^TM^ concentrations were 50× and its dilutions to 30× and 10× and treatment was used during infection, post infection and a combination of during and post infection. Based on those standardizations, infections in primary cells were performed using MOI 1 for 24 h for all subsequent experiments. Treatment with BIO-PLY^TM^ was applied during and post-infection using the 50× concentration.

### 2.6. TCID50 Assay

The antiviral activity of BIO-PLY^TM^ was assessed by measuring the number of infectious viral particles via TCID50 assay and the amount of viral RNA via RT-qPCR in PAM and PIM. TCID50 assays were performed as previously described [48]. Infection and treatment layout was used as described in Section 2.5. Briefly, macrophages (both PAM and PIM) were infected with either PRRSV NC-174, NC-134 or VR2332 at MOI 1 for 1 h with or without BIO-PLY^TM^ at a 1:2 ratio with media. Then, inoculum was removed and substituted by RPMI with or without BIO-PLY^TM^ for 24 h. This way, BIO-PLY^TM^ was added during and post-infection. Supernatants of infected macrophages (both PAM and PIM) were collected at 24 h and used to perform the TCID50 assay on a confluent 96 well plate of MA-104 cells. Cell fixation, staining with crystal violet and calculations were performed at seven days post infection as previously described [48].

### 2.7. RT-qPCR

Macrophages were lysed and RNA was extracted using the PureLink™ RNA Micro Scale Kit (Catalog number 12183016, Invitrogen, Waltham, MA, USA). cDNA synthesis was performed using iScript™ cDNA Synthesis Kit (Biorad, Hercules, CA, USA).

The primers used for the detection of the genomic copies of all three PRRSV strains were nsp9 F (5′-CCTGCAATTGTCCGCTGGTTTG-3′) and nsp9 R (5′-GACGACAGGCCACCTCTCTTAG-3′), previously described by Spear and Faaberg (2015) [49].

Details of the primers used to detect cytokine expression are listed in Table 1. Primers were purchased from Integrated DNA Technologies. The cytokines analyzed for this study were IL-1β, IL-4, IL-6, IL-10, IL-12 and TNF-α and RPS-24 as housekeeping gene. To reduce the variation between samples in the RT-qPCR raw data, the Ct values of the housekeeping gene were used to perform a ΔCt normalization. Subsequent normalizations to mock or virus were performed depending on the specific experiment [50].

### 2.8. Cytokine Production

Cytokine production of infected macrophages was additionally measured via multiplex. After infection, supernatants were collected at 24 h and a MILLIPLEX^®^ Porcine Cytokine and Chemokine Magnetic Bead Panel was used (kit includes GM-CSF, IFN-γ, IL-1α, IL-1β, IL-1ra, IL-2, IL-4, IL-6, IL-8, IL-10, IL-12, IL-18, TNF-α) following manufacturer instructions (Catalog number PCYTMG-23K, Millipore Sigma, Burlington, MA, USA).

### 2.9. ROS Production

To measure ROS production, PAM and PIM were seeded in black-wall 96 well plates (Greiner Bio-one, Monroe, NC, USA) at a density of 100,000 cells per well. Cells were infected at MOI 1 and treated with BIO-PLY^TM^ as described in Section 2.5. After incubation for 24 h, infected cells were washed with fresh media, and Dihydrorhodamine 123 (or DHR-123) (Chemodex, St. Gallen, Switzerland) was added into the wells at a concentration of 10 μM following manufacturer’s instructions. Plates were incubated for 15 min at 37 °C in a 5% CO_2_ incubator protected from light, and fluorescence was measured using a spectrophotometer plate reader (BioTek, Winooski, VT, USA) at different timepoints (0 h 6 h, 12 h, 24 h, 36 h and 48 h).

### 2.10. Cytotoxicity

BIO-PLY^TM^ cytotoxicity in cell lines, PAM and PIM was measured using the CellTox^TM^ Green Cytotoxicity Assay kit (Promega, Madison, WI, USA) following manufacturer’s instructions. Briefly, BIO-PLY^TM^ was diluted in RPMIc at different concentrations and then incubated with newborn porcine tracheal cells (NPTr) [52], MA-104 (ATCC, CRL-2378.1T), PAM or PIM for 24 h or 48 h at 37 °C in a 5% CO_2_ incubator protected from light. Then, the fluorescent dye was added to the cells for subsequent measurement of fluorescence as an indicator of cellular death. Measurements were done using a spectrophotometer plate reader at an excitation wavelength of 485–500 nm and emission of 520–530 nm (BioTek, Winooski, VT, USA) and results are shown as relative fluorescence units (RFU).

### 2.11. Flow Cytometry

Cell viability data was acquired via flow cytometry using CytoFLEX flow cytometer (Beckman Coulter, Pasadena, CA, USA). Briefly, PAM and PIM were isolated and treated with BIO-PLY^TM^ or PBS in media as previously described. After incubation, cells were resuspended in buffer composed of PBS supplemented with 2% Fetal Bovine Serum (FBS) and EDTA 2 mM (Sigma-Aldrich, St. Louis, MO, USA), and stained with primary and secondary antibodies for 1 h each. The primary antibodies used were a mouse anti-pig CD163 PE-conjugated antibody (MA5-16476, Invitrogen, Waltham, MA, USA) and a mouse anti-pig CD-172a (IgG1, MCA2312GA, clone BL1H7, BioRad, Hercules, CA, USA). Then, a polyclonal goat anti-mouse IgG1 BV421-conjugated secondary antibody (, 115-675-205, Jackson ImmunoResearch, West Grove, PA, USA). Cells were stained with DAPI (Sigma-Aldrich, St. Louis, MO, USA) following manufacturer’s instruction before flow cytometry analysis. Data analysis was performed with FlowJo version 10.5.3 (FlowJo LLC, BD Life Science, Ashland, OR, USA).

## 3. Results

When we compared the two isolation methods for the adherent and non-adherent MNP described in Section 2.2, there were no significant differences between the two processes (magnetic beads vs. cell attachment) (Appendix A) and thus, the isolation of macrophages for subsequent analyses was performed through adherence.

Based on the standardizations described in Section 2.5, infections in primary cells were performed using MOI 1 for 24 h for all subsequent experiments. Treatment with BIO-PLY^TM^ was applied during and post-infection using the 50× concentration.

### 3.1. Antiviral Capacity of BIO-PLY^TM^

When porcine BIO-PLY^TM^-treated and non-treated macrophages were compared, we observed a statistically significant reduction in the genome copies measured by RT-qPCR that exceeded an 80% reduction in infected PAM (Figure 2, left) and a 70% reduction in infected PIM (Figure 2, right). This effect was also observed using equine BIO-PLY^TM^ in PAM and PIM (Appendix A).

When measuring the reduction in infectious viral particles via TCID50 assay, there was also a statistically significant difference between BIO-PLY^TM^-treated and non-treated cells, with an over a 90% reduction for both PAM (Figure 3, left) and PIM (Figure 2, right).

### 3.2. Cytokine Modulation

We did not find any BIO-PLY^TM^ -triggered differences in cytokine expression in PAM by RT-qPCR; however, statistical differences were observed in some cases when we measured protein levels by a multiplex magnetic bead assay (Figure 4). PAM treated with BIO-PLY^TM^ produced higher amounts of interleukin-1 beta (IL-1β), but there was no statistically significant difference between its production in BIO-PLY^TM^ treated and non-treated infected cells (Figure 4A). In the case of interleukin-6 (IL-6), we observed significant differences triggered by BIO-PLY^TM^ in non-infected and PRRSV NC-134 infected macrophages, but not for PRRSV NC-174 (Figure 4C). Similarly, interleukin-12 (IL-12) showed a significant difference in PRRSV NC-134 infected PAM, but not in non-infected or PRRSV NC-174 infected cells (Figure 4E). BIO-PLY^TM^ significantly decreased TNF-α production in non-infected and PRRSV NC-174-infected PAM, but not in cells infected with PRRSV NC-134 (Figure 4F).

In the case of PIM, RT-qPCR also showed no significant differences in cytokine expression between BIO-PLY^TM^ treatment and untreated PIM (Figure 5). We observed statistically significant differences in the production of and IL-1β protein in non-infected PIM, and PRRSV NC-134 infected PIM, but not in PRRSV NC-174 infected cells (Figure 5A). Additionally, we observed statistically significant differences in production of interleukin-4 (IL-4) (Figure 5B), interleukin-10 (IL-10) (Figure 5D) and TNF-α (Figure 5F) between BIO-PLY^TM^-treated and mock, but no differences between BIO-PLY^TM^ treated and untreated infected cells with either PRRSV NC-134 or PRRSV NC-174.

### 3.3. Reactive Oxygen Species (ROS) Production

We observed different patterns of reactive oxygen species (ROS) production stimulated by each PRRSV strain (Figure 6). The observed ROS values for non-infected cells show the baseline or background for macrophage ROS production in culture. The low pathogenic PRRSV NC-134 triggered an impairment in ROS production of infected macrophages, which was restored by the treatment with BIO-PLY^TM^ (Figure 6A). In the case of the highly pathogenic NC-174, infection led to an increased ROS production that was brought back to the levels of non-infected cells with the application of BIO-PLY^TM^ (Figure 6B). This pattern was also observed in VR-2332 (Figure 6C). Same patterns were observed in PAM and PIM (data not shown).

### 3.4. BIO-PLY^TM^ Cytotoxicity and Cell Viability

BIO-PLY^TM^ cytotoxicity was measured in NPTr (Figure 7) and MA-104 (data not shown) cell lines with the same results for swine and equine BIO-PLY^TM^. High levels of mortality correlate with increasing concentrations of BIO-PLY^TM^, particularly from 70% to 100% (Figure 7A). BIO-PLY^TM^ supernatants did not cause any cytopathic effect in either cell line when TCID50 assays were performed (data not shown). In the case of macrophages, both PAM and PIM showed no increase in mortality via flow cytometry when incubated with BIO-PLY^TM^ at 50% for 24 h (Figure 7B) or upon PRRSV-2 infection with or without BIOPLY (data not shown). Gate strategy available in Appendix A. When measured via cytotoxicity assay for 48 h (Figure 7C), swine BIO-PLY^TM^ seemed to cause slightly higher levels of mortality in both PAM and PIM compared to mock. However, most of the differences between treatment and mock were not statistically significant. Finally, it is important to note that the primary cells used for the assay went through a long isolation process (described in Section 2.2) and were frozen before the assay, making them more fragile than the cell lines.

## 4. Discussion

BIO-PLY^TM^ was previously evaluated and shown to be effective as an antibacterial treatment for different types of both Gram-positive and Gram-negative bacteria [32,47]. In this study, for the first time, we evaluated its antiviral potential against different RNA viruses in vitro. We have assessed BIO-PLY^TM^ antiviral activity against PRRSV-2 in lung primary macrophages and its potential capacity to modulate macrophages’ function and immune responses.

The capacity of BIO-PLY^TM^ to control viral infection was observed not only by a reduction on viral copies via RT-qPCR, but also by a decrease in the production of infectious viral particles measured by TCID50 method. In this study we tested the effects of BIO-PLY^TM^ in cells coming from two different compartments of the lung, PAM obtained from the alveoli and PIM retrieved from the parenchyma, which have been suggested to play different roles during PRRSV infection [53]. Future studies need to be performed to evaluate BIO-PLY^TM^ for its antiviral capacity in vivo.

We did not observe any relevant altered patterns of cytokine regulation in PAM or PIM triggered by BIO-PLY^TM^ treatment. Therefore, we can only speculate that the antiviral effects of BIO-PLY^TM^ seem to be on the virus itself or on an alteration of its cycle rather than on the immunomodulation of macrophages. This is similar to other drugs that do not inactivate their target pathogens but rather inhibit their replication [54] such as Acyclovir [55,56,57,58] or Remdesivir [54,59,60] among others. We also measured the antiviral effect of BIO-PLY^TM^ in two additional human RNA viruses, influenza A viruses and coronavirus (details of infection described in Appendix A). In the case of influenza virus, we observed a significant decrease on the viral copies detected via RT-qPCR (Appendix A). In the case of human coronavirus, we observed a reduction in the viral titer using TCID50 assay (Appendix A).

These results not only demonstrate the efficacy of BIO-PLY^TM^ against additional RNA viruses, but also its potential inter-species use, since these experiments showed the effect of swine BIO-PLY^TM^ in monkey and canine cell lines (Figure 7 and Appendix A), and a human cell line used for coronavirus infection (Huh-7) (Appendix A) without cytotoxicity. This is possible because BIO-PLY^TM^ is the acellular fraction of PRP, allowing its allogenic use. For that reason, we also tested the effect of equine BIO-PLY^TM^ in primary swine cells (PAM and PIM) following the same methods described for swine BIO-PLY^TM^ and observed comparable levels of viral reduction via TCID50 (Appendix A). Additionally, the lack of cytotoxicity was previous supported in vivo by a clinical trial performed in COVID-19 patients [37]. The phase I/II study showed that autologous activated PRP used intravenously was safe and was not associated with serious or adverse events [37].

In relation to the immunomodulatory effects of BIO-PLY^TM^ in infected macrophages, the biologic did not show any major differences in cytokine production on BIO-PLY^TM^ treated compared to non-treated cells. However, there were differences in ROS production triggered by each PRRSV strain, where the low pathogenic strain NC-134 showed an impairment in ROS production while the high pathogenic NC-174 led to an increasing pattern over time. Previous literature has linked ROS production impairment with the dysregulation of antioxidant enzymes during viral infections, particularly in PRRSV, where the combination of an excessive concentration of ROS and the malfunction of antioxidant agents increases the effects of oxidative stress within cells [61,62,63]. We hypothesized that ROS levels are restored to their baseline because of the reduction in viral load and replication during BIO-PLY^TM^ treatment: the drop in virus cargo would reduce ROS modulation. Therefore, this ability of BIO-PLY^TM^ to restore ROS production in PRRSV-infected cells by bringing it back to mock values could be one of the key mechanisms through which this particular biologic reduces the impact of oxidative stress during PRRSV infections in vitro, and should be further analyzed.

Some limitations of the studies described here need to be considered. The main one is that even though our ex vivo methodology aims to identify cell responses that are similar to what would happen in vivo, when immune cells are isolated, possible synergistic effects and responses that depend on interactions with other cell types are overlooked; therefore, even though the results of this study provide an important stepping stone for the potential use of BIO-PLY^TM^ as an antiviral therapeutic, there is a need for the further development of in vivo studies to confirm the absence of systemic effects and to assess its efficacy. Another limitation to consider relies on the processing method of BIO-PLY^TM^. Since it is a biologic, considerable lot-to-lot variability is likely. However, the pooling batches of BIO-PLY^TM^ mitigates this variability which was observed since different BIO-PLY^TM^ lots (*n* = 3) yielded similar results. Lastly, it is important to note that the primary porcine lung cells used for this study were processed, frozen, and stored.

Finally, BIO-PLY^TM^ can be considered a potential candidate to treat and control polymicrobial diseases, since the same compound and concentration showed both antibacterial and antiviral properties. Future directions for the use of BIO-PLY^TM^ as an antimicrobial agent will move towards the treatment of combined viral and bacterial polymicrobial diseases in vivo, especially towards swine respiratory disease complex and human pneumonia. Additionally, since BIO-PLY^TM^ is a poly-peptide biologic [27,33], identifying the specific peptides that are more involved in the development of the antimicrobial responses is crucial since synthetized peptides are easier to produce than biologic-based treatments.

## 5. Conclusions

BIO-PLY^TM^ presents an effective antiviral treatment for PRRSV-2 infection in vitro, as well as against other RNA viruses such as human A influenza viruses and coronavirus when used both intra and inter-species. This study is the steppingstone for evaluating future potential applications of BIO-PLY^TM^ as an antiviral treatment in vivo. The antiviral activity of BIO-PLY^TM^ combined with its previously described antibacterial activity shows great potential for BIO-PLY^TM^ as a unique agent to reduce the burden of polymicrobial diseases.

## 6. Patents

Two invention disclosures have been submitted for the antiviral capacity of BIO-PLY^TM^ (NCSU 20186, D2023-0056). A future patent will be filed on those properties.

## Figures and Tables

**Figure 1 viruses-14-02666-f001:**
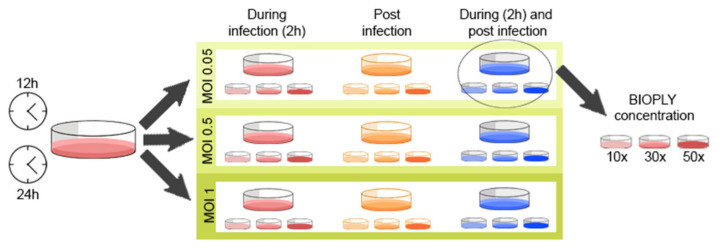
Standardization process for the infection settings and BIO-PLY^TM^ treatment, considering infection time (12 h or 24 h), MOI (1, 0.5 or 0.05), BIO-PLY^TM^ treatment (during, post or during and post infection) and BIO-PLY^TM^ concentration (10×, 30× or 50×).

**Figure 2 viruses-14-02666-f002:**
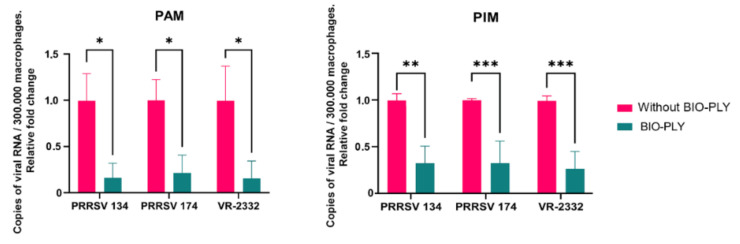
Effect of BIO-PLY^TM^ treatment on PRRSV infection of porcine alveolar macrophages (PAM) and pulmonary intravascular macrophages (PIM). Briefly, PAM and PIM were infected with PRRSV with and without BIO-PLY^TM^ treatment, and viral copies produced after 24 h of infection were measured via RT-qPCR. All experiments were performed using technical duplicates in 3 biological replicates. Obtained values were normalized to the average values for each of the three PRRSV strains without BIO-PLY treatment (set as 1) to reduce variation between biological samples and visualized as proportions based on the average values of infection. Results were analyzed via two-way ANOVA and displayed showing means and standard error of mean (SEM). *p*-values were expressed using asterisks following APA and Prism standards. * *p* ≤ 0.05, ** *p* ≤ 0.01, *** *p* ≤ 0.001.

**Figure 3 viruses-14-02666-f003:**
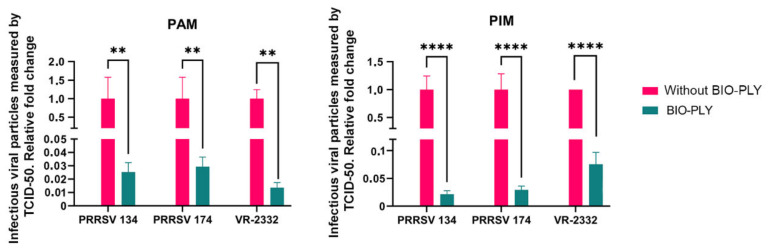
Effect of BIO-PLY^TM^ treatment on the production of infectious viral particles in porcine alveolar macrophages (PAM) and pulmonary intravascular macrophages (PIM) measured as TCID50/mL. Briefly, PAM and PIM were infected with PRRSV with and without BIO-PLY^TM^ treatment. Supernatants were collected after 24 h of infection and used to perform TCID50 assays. All experiments were performed using technical duplicates in 3 biological replicates. Obtained values were normalized to the mean of the virus values without BIO-PLY treatment (set as 1) to reduce variation between biological samples and visualized as proportions based on the average values of infection. Results were analyzed via two-way ANOVA and displayed showing means and standard error of mean (SEM). *p*-values were expressed using asterisks following APA and Prism standards. ** *p* ≤ 0.01, **** *p* ≤ 0.0001.

**Figure 4 viruses-14-02666-f004:**
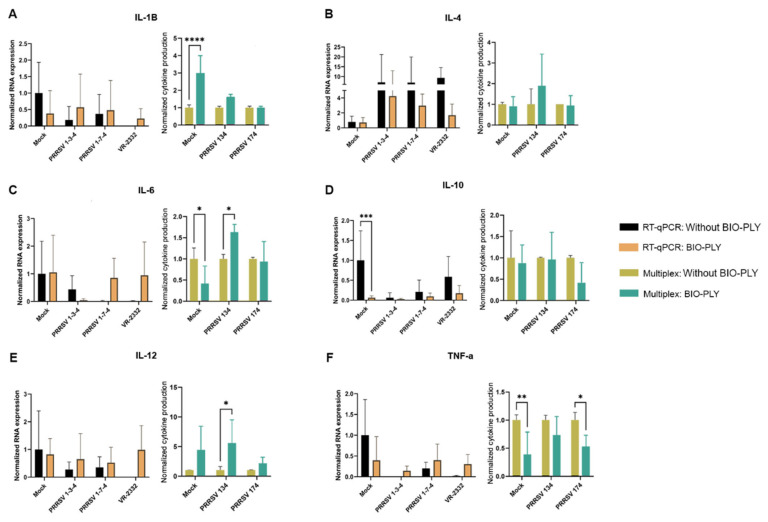
Effect of BIO-PLY^TM^ on cytokines (mRNA or protein) in porcine alveolar macrophages (PAM) infected with PRRSV. Briefly, cells were infected with PRRSV for 24 h; after infection, cytokine expression was measured via RT-qPCR (shown in the left graph of each panel) and cytokine production in supernatants via multiplex (shown in the right graph of each panel). (**A**) IL1-β, (**B**) IL-4, (**C**) IL-6, (**D**) IL-10, (**E**) IL-12, (**F**) TNF-α. All experiments were performed using technical duplicates in biological replicates ranging from 3 to 5 pigs per experiment. Obtained values were normalized to the average of the mock values. Results were analyzed via two-way ANOVA and displayed showing means and standard error of mean (SEM). *p*-values were expressed using asterisks following APA and Prism standards. * *p* ≤ 0.05, ** *p* ≤ 0.01, *** *p* ≤ 0.001, **** *p* ≤ 0.0001.

**Figure 5 viruses-14-02666-f005:**
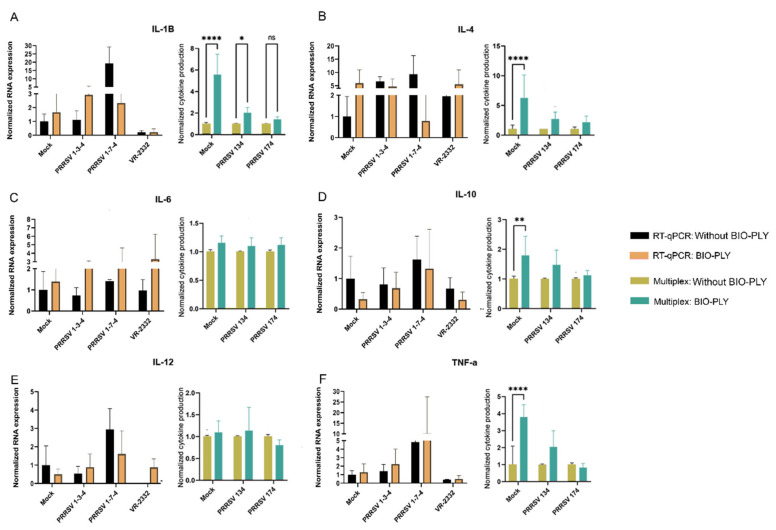
Immunomodulatory capacity of BIO-PLY^TM^ on porcine pulmonary intravascular macrophages (PIM) infected with PRRSV. Briefly, cells were infected with PRRSV for 24 h; after infection, cytokine expression was measured via RT-qPCR (shown in the left graph of each panel) and cytokine production in supernatants via multiplex (shown in the right graph of each panel). (**A**) IL1-β, (**B**) IL-4, (**C**) IL-6, (**D**) IL-10, (**E**) IL-12, (**F**) TNF-α.All experiments were performed using technical duplicates in biological replicates ranging from 3 to 5 pigs per experiment. Obtained values were normalized to the average of the mocks. Results were analyzed via two-way ANOVA and displayed showing means and standard error of mean (SEM). *p*-values were expressed using asterisks following APA and Prism standards. * *p* ≤ 0.05, ** *p* ≤ 0.01, **** *p* ≤ 0.0001.

**Figure 6 viruses-14-02666-f006:**
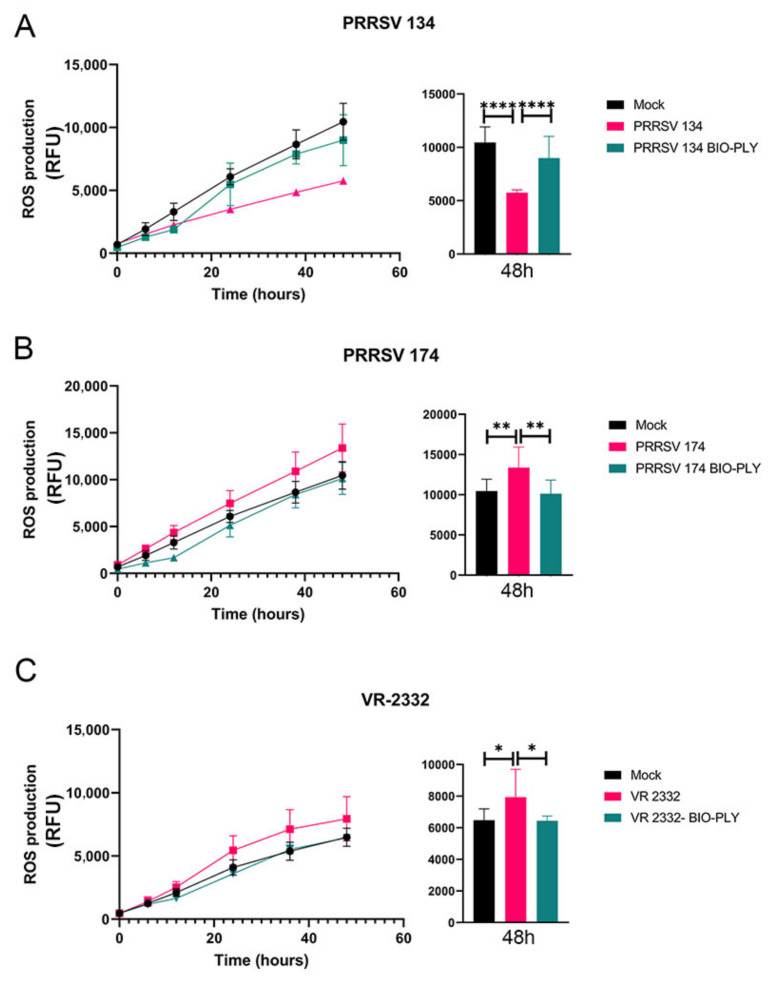
Effect of BIO-PLY^TM^ treatment on the ROS production of porcine alveolar macrophages (PAM) infected with PRRSV-2. Briefly, cells were infected with (**A**) PRRSV NC-134, (**B**) PRRSV NC-174, or (**C**) PRRSV VR-2332, or left in culture alone (mock). ROS production was measured at different time points (left) and visualized at 48 h (right). All experiments were performed using technical duplicates in 4 biological replicates. Obtained values were normalized to the average of the mocks. Results were analyzed via ANOVA and displayed showing means and standard error of mean (SEM). *p*-values were expressed using asterisks following APA and Prism standards. * *p* ≤ 0.05, ** *p* ≤ 0.01, **** *p* ≤ 0.0001.

**Figure 7 viruses-14-02666-f007:**
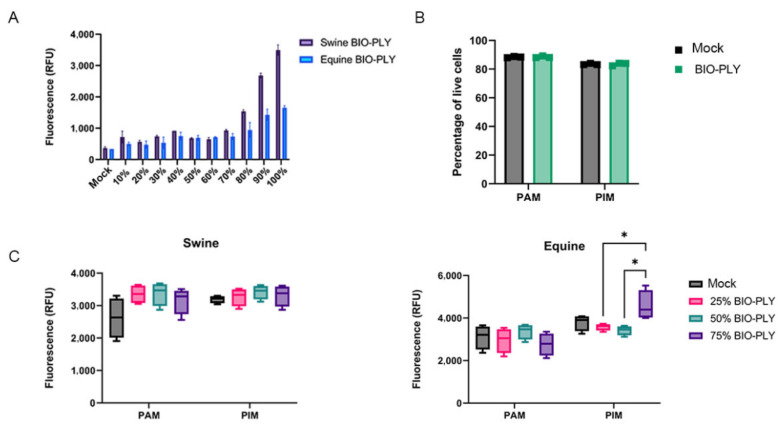
Cell viability and BIO-PLY^TM^ cytotoxicity in cell line and primary macrophages PAM and PIM. (**A**) Effect of different BIO-PLY^TM^ concentrations in NPTr cell line using the Promega CellTox green cytotoxicity assay. (**B**) Effect on the viability of PAM and PIM treated with 50% swine BIO-PLY^TM^ for 24 h measured via flow cytometry and LIVE/DEAD staining. (**C**) The impact of swine (left) and equine (right) BIO-PLY^TM^ treatment for 48 h in PAM and PIM using the same Promega kit. All experiments were performed using technical duplicates in 4 biological replicates. Results are expressed as relative fluorescence units (RFU) and were analyzed via two-way ANOVA and displayed showing means and standard error of mean (SEM). *p*-values were expressed using asterisks following APA and Prism standards. * *p* ≤ 0.05.

**Table 1 viruses-14-02666-t001:** Summary of primers used for cytokine detection via RT-qPCR.

Cytokine	Primer	Sequence	Reference
IL-1β	Forward	5′-TGC CAA CGT GCA GTC TAT GG-3′	[51]
	Reverse	5′-TGG GCC AGC CAG CAC TAG-3′	
IL-4	Forward	5′-GCC GGG CCT CGA CTG T-3′	[42]
	Reverse	5′-TCC GCT CAG GAG GCT CTT C-3′	
IL-6	Forward	5′-CTG CTT CTG GTG ATG GCT ACT G-3′	[42]
	Reverse	5′-GGC ATC ACC TTT GGC ATC TT-3′	
IL-10	Forward	5′-GAG CCA ACT GCA GCT TCC A-3′	[42]
	Reverse	5′-TCA GGA CAA ATA GCC CAC TAG CTT-3’	
IL-12	Forward	5′-GGA GCA CCC CAC ATT CCT ACT-3′	[42]
	Reverse	5′-TTC TCT TTT GTT CTT GCC CTG AA-3′	
TNF-α	Forward	5′-TGG TGG TGC CGA CAG ATG -3′	[42]
	Reverse	5′-CAG CCT TGG CCC CTG AA -3′	
RPS24	Forward	5′-TTT GCC AGC ACC AAC GTT G-3′	[42]
	Reverse	5′-AAG GAA CGC AAG AAC AGA ATG AA-3′	

## Data Availability

Not applicable.

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
