# Peer review of "Effect of BIO-PLYTM, a Platelet-Rich Plasma Derived Biologic on PRRSV-2-Infected Macrophages"

_viruses, 2022, doi:10.3390/v14122666_

Round 1
Reviewer 1 Report
Frias-De Diego et al., evaluated the effects of a novel platelet-rich plasm-derived biologic, "BIO-PLY," from swine and equine origin. The authors expect the "BIO-PLY" product to inhibit the in vitro infection of two RNA viruses: porcine reproductive and respiratory syndrome virus and influenza virus. They used alveolar macrophages and evaluated the viral loads, cytokine and reactive oxygen species production, and viability. The manuscript is straightforward, easy to follow, well-structured, engaging, and worthy of publication in Viruses after some issues are resolved.
I recommend the authors use the Ct value to express the values as the copies of viral RNA/number of PAM or PIM. The use of "percentage of viral copies" is confusing and inappropriate. Same for figure 2 and 3
Did the authors evaluate the viability of infected cells (Figure 7)? It will be interesting to confirm that the reduction in the viral loads is not an effect of the loss of cell viability, more than an antiviral effect.
Lines 362-365: additional experiments are needed to support this statement; otherwise, modify accordingly.
Lines 372-374: The authors must provide additional experiments such as Western-Blot or Norther-Blot to support this statement; otherwise, modify accordingly.
Author Response
Frias-De Diego et al., evaluated the effects of a novel platelet-rich plasm-derived biologic, "BIO-PLY," from swine and equine origin. The authors expect the "BIO-PLY" product to inhibit the in vitro infection of two RNA viruses: porcine reproductive and respiratory syndrome virus and influenza virus. They used alveolar macrophages and evaluated the viral loads, cytokine and reactive oxygen species production, and viability. The manuscript is straightforward, easy to follow, well-structured, engaging, and worthy of publication in Viruses after some issues are resolved.
I recommend the authors use the Ct value to express the values as the copies of viral RNA/number of PAM or PIM. The use of "percentage of viral copies" is confusing and inappropriate. Same for figure 2 and 3
We followed the reviewer’s suggestion, and we changed the figures, legends, and text accordingly. We have visualized Ct values normalizing to the house keeping gene (ΔCt) and then performed a ratio with the mean of the virus alone set as 1. We think that this visualization of the data is easier to understand; this type of visualization or the use of the % has been previously used in other recent publications (e.i. Zhang et al 2021 Cell Research (2022) 32:9–23 Fig 4b; https://doi.org/10.1038/s41422-021-00581-y, Hui et al Antiviral Res. 2022 Nov 16;105465. doi: 10.1016/j.antiviral.2022.105465). Additionally, this type of normalization allows to reduce the high variability when using primary macrophages isolated from different animals.
Did the authors evaluate the viability of infected cells (Figure 7)? It will be interesting to confirm that the reduction in the viral loads is not an effect of the loss of cell viability, more than an antiviral effect.
We thank the reviewer for this valuable comment. We have tested viability of macrophages infected or not with PRRSV-2 and treated or not with BIO-PLYTM. In our experimental layout we did not see any significant difference in cell viability between all the conditions evaluated by flow cytometry. We have added a figure 1 of the data below. The findings are included in the text (lines 341-342) as data not shown. If the reviewer thinks that data need to be included in the manuscript as a supplementary figure, we will be happy to do so.
Lines 362-365: additional experiments are needed to support this statement; otherwise, modify accordingly.
During the revision we have performed and internalization and binding assay with BIO-PLYTM and PRRSV-2; unfortunately data did not show any definitive conclusion, therefore we have modified the text accordingly (lines 368-370).
Lines 372-374: The authors must provide additional experiments such as Western-Blot or Norther-Blot to support this statement; otherwise, modify accordingly.
We have modified the text accordingly (lines 378-380).

Reviewer 2 Report
This manuscript described the in vitro antiviral effect of BIO-PLYTM from both swine and equine origin against PRRSV and other RNA viruses. The replication of three PRRSV strains was significantly inhibited by BIO-PLYTM based virus load reductions determined by RT-qPCR and virus titration. The effect of BIO-PLYTM on cytokine production suggested that BIO-PLYTM may have a direct effect on PRRSV. Furthermore, BIO-PLYTM treatment restored the ROS production in macrophages infected with PRRSV. Overall, it is very interesting research on a new control method for RNA viruses, and the manuscript is well organized. I only have few comments.
Based on the results presented in this study, BIO-PLYTM treatment always bring the ROS production back to mock values no matter it is upregulated or downregulated by viral infection. I am curious about the mechanism. Could you add more information about this in the discussion section?
In line 380, BIO-PLYTM treatment significantly decreased the viral genome copy number of influenza virus but not the viral titer. Could you please explain this conflict?
Author Response
This manuscript described the in vitro antiviral effect of BIO-PLYTM from both swine and equine origin against PRRSV and other RNA viruses. The replication of three PRRSV strains was significantly inhibited by BIO-PLYTM based virus load reductions determined by RT-qPCR and virus titration. The effect of BIO-PLYTM on cytokine production suggested that BIO-PLYTM may have a direct effect on PRRSV. Furthermore, BIO-PLYTM treatment restored the ROS production in macrophages infected with PRRSV. Overall, it is very interesting research on a new control method for RNA viruses, and the manuscript is well organized. I only have few comments.
Based on the results presented in this study, BIO-PLYTM treatment always bring the ROS production back to mock values no matter it is upregulated or downregulated by viral infection. I am curious about the mechanism. Could you add more information about this in the discussion section?
We thank the reviewer for this question, and we have included more information in the discussion (lines 409-412). We hypothesize that ROS levels are restored to their baseline because of the reduction in viral load and replication during BIO-PLYTM treatment: the drop in virus cargo would reduce ROS modulation.
In line 380, BIO-PLYTM treatment significantly decreased the viral genome copy number of influenza virus but not the viral titer. Could you please explain this conflict?
We think that the conflict between RT-qPCR and TCID50 could be related to the different sensitivity of the two methods. We performed TCID50 method using 10-fold dilution system therefore we might have not been able to see small changes in infectious viral particles levels. A new test with 2-fold dilution would potentially help clarifying this aspect but we will remove the sentence in the text to avoid any conflict. Future studies will be performed to clarify this discrepancy and to confirm the TCID50 outcome for influenza A viruses.